# A False Sense of Privacy: Evaluating Textual Data Sanitization Beyond Surface-level Privacy Leakage

## Abstract

Sanitizing sensitive text data for release often relies on methods that remove personally identifiable information (PII) or generate synthetic data. However, evaluations of these methods have focused on measuring surface-level privacy leakage (e.g., revealing explicit identifiers like names). We propose the first semantic privacy evaluation framework for sanitized textual datasets, leveraging re-identification attacks. On medical records and chatbot dialogue datasets, we demonstrate that seemingly innocuous auxiliary information, such as a mention of specific speech patterns, can be used to deduce sensitive attributes like age or substance use history. PII removal techniques make only surface-level textual manipulations: e.g., the industry-standard Azure PII removal tool fails to protect 89% of the original information. On the other hand, synthesizing data with differential privacy protects sensitive information but garbles the data, rendering it much less useful for downstream tasks. Our findings reveal that current data sanitization methods create a *false sense of privacy*, and underscore the urgent need for more robust methods that both protect privacy and preserve utility.

## 1 Introduction

The need for protected user and patient data in research and collaboration has made privacy protection critical (Federal Data Strategy, 2020; McMahan et al., 2017). Organizations handling personal identifiers, location traces, and behavioral patterns face risks when adversaries can link multiple datasets to re-identify individuals or infer sensitive attributes from released data. To mitigate sensitive information disclosure risks, two sanitization approaches are widely used (Garfinkel, 2015): (1) removing explicit identifiers and (2) generating synthetic datasets that mimic the statistical properties of the original data. Explicit identifier removal often redacts sensitive information by lexical matching; data synthesis produces new generations that are not considered to contain real units from the original data (Stadler et al., 2022; Rankin et al., 2020). While these methods eliminate direct identifiers and modify data at the surface level, they may fail to address subtle semantic cues that could compromise privacy. Additionally, sanitization methods must also maintain utility while providing privacy protection. This leads to a critical question: *Do these methods truly protect data, or do they provide a false sense of privacy?*

To adequately address privacy risks in data sharing, sanitization methods must be evaluated through *semantic*-level analysis under realistic threat models. Consider a sanitized medical dataset containing Alice's de-identified record (Figure 1). An adversary aware of some auxiliary information about Alice (e.g., drinks one glass of wine daily), through sources such as social media, could exploit similarities between this external information and entries in the sanitized dataset (Ganta et al., 2008) and re-identify Alice's record. Semantic matching could then expose sensitive attributes like her mental state or substance consumption, even if the sanitized dataset exhibits minimal literal overlap with the adversary's prior information. For example, in Alice's case, the sanitization method might generalize Alice's specific marijuana use into a broader category of substance use. If we compare only explicit identifier, we would report no leakage, as the text do not match precisely.

However, conventional privacy evaluation methods often rely on pattern matching using a fixed dictionary and removal of direct identifiers like names, deeming data safe when no matches are

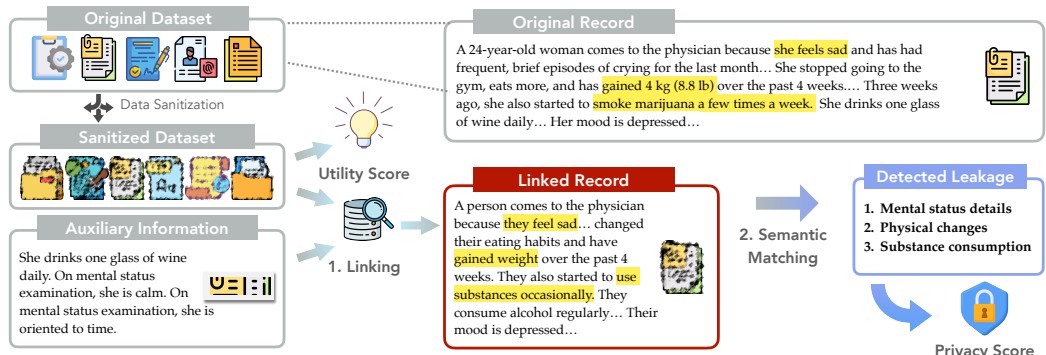

Figure 1: Our privacy evaluation framework overview: First, in the **linking** stage, we use innocuous auxiliary information to find potential matches in the sanitized dataset using a sparse retriever. Second, in the **semantic matching** stage, we *semantically* analyze the matched records to identify information leakage. The framework calculates both a privacy score based on the detected information leakage and a utility score to measure the practical value of the sanitized dataset.

found (Pilán et al., 2022). This practice ignores the fact that privacy risks extend beyond these explicit identifiers to quasi-identifiers–seemingly innocuous information that, when combined, can reveal sensitive details (Sweeney, 2000; Weggenmann & Kerschbaum, 2018). Therefore, privacy evaluation must consider beyond exact text matching to semantic matching between original and sanitized data.

To address this gap in evaluation and provide an effective privacy measurement, we introduce the first framework that quantifies the amount of detail inferrable about an individual from sanitized data, given auxiliary information (Ganta et al., 2008), while also evaluating data utility to assess the privacy-utility trade-off. Grounded in statistical disclosure control (SDC) guidelines used by the US Census Bureau for anonymizing tabular data (Abowd et al., 2023), which use reconstruction attacks to evaluate data sanitization, our two-stage process (Figure 1) adapts these principles to unstructured text. The first stage, **linking**, employs a sparse retriever to match the given auxiliary information with de-identified, sanitized records that may contain additional sensitive information. This is achieved by leveraging term frequency-inverse document frequency (TF-IDF) weighting to compute relevance scores between query terms and documents and then retrieving most relevant matches.

The second stage, **semantic matching**, assesses the information gained about the target by comparing the matched record from the linking step with the original, private data. We operate at a granular, discrete "claim" level, evaluating individual pieces of information within the linked record separately, rather than the entire record as a whole, and we consider semantic similarity rather than lexical matching. This allows for a more nuanced assessment of privacy risks.

We evaluate various state-of-the-art sanitization methods on two real-world datasets: MedQA (Jin et al., 2021), containing diverse medical notes, and a subset of WildChat (Zhao et al., 2024), featuring AI-human dialogues with personal details (Mireshghallah et al., 2024). Specifically, we compare the two categories of sanitization approaches discussed above: (1) identifier removal techniques, including commercial PII removal, LLM-based anonymizers (Staab et al., 2024), and sensitive span detection (Dou et al., 2024); and (2) data synthesis methods using GPT-2 fine-tuned on private data, with and without differential privacy (Yue et al., 2023). We assess both privacy and utility, measuring leakage with our metric and lexical matching, and evaluating sanitized datasets on domain-specific downstream tasks to investigate the privacy-utility tradeoff in different data sanitization method. For example, on the MedQA medical question-answering task, utility of the sanitized data is measured by the task accuracy, as datasets with higher utility should preserve more useful information for the model to correctly answer the question.

*Our main finding is that current dataset sanitization methods for text data often provide a false sense of privacy*. Specifically: (1) State-of-the-art PII removal methods are surface-level and still exhibit significant leakage, with 89% of original information still inferable when providing the attacker with

access to auxiliary information and the ability to make partial attribute matches. (2) Without differential privacy, synthesized data still exhibits leakage (with 55% of the information re-identifiable). (3) Differentially private (DP) synthesis methods provide the strongest privacy protections but can significantly reduce utility, particularly for complex tasks. Our experiments on the medical question-answering benchmark (MedQA) show a $-4\%$ decrease in performance compared to the degenerate baseline, where we sanitize by removing all information. These results suggest that data generated through DP synthesis methods actively diminishes task performance. DP synthesis also degrades the textual quality on the synthesized documents by 36% when measuring on a 1 to 5 Likert scale using a language model. We conduct comprehensive ablations, including using different semantic matching techniques and changing the type of auxiliary information used for de-identification. Our results highlight the necessity to develop methods protecting privacy that go beyond surface-level protections and obvious identifiers, ensuring a more comprehensive approach to data privacy in text-based domains.

## 2 PRIVACY METRIC

As shown in Figure 1, given a sanitized dataset, our framework employs a linking step and a semantic similarity match to evaluate the privacy protection ability of the sanitizer.

**Problem statement.** Let $\mathcal{D}_{\text{original}} = \{x^{(i)}\}_{i=1}^{N}$ denote the original dataset and $\mathcal{D}_{\text{sanitized}} = S(\mathcal{D}_{\text{original}}) = \{y^{(i)}\}_{i=1}^{M}$ the sanitized dataset for the given data sanitization method of interest $S$.

Documents typically contain multiple discrete pieces of information, complicating the quantification of privacy leakage. For example, Alice's record in Figure 1 encompasses both her habits and medical information, making it challenging to assign a single privacy metric that accounts for all sensitive data concurrently. To address this issue and facilitate a more fine-grained approach to privacy evaluation, we atomize the data records. Adopting the core concept introduced by Min et al. (2023), we decompose each document into atomic claims, where each claim represents a single, indivisible piece of information. In our framework, we partition each data record $x^{(i)}$ into a set of atomized claims $x_j^{(i)}$.

Our goal is to evaluate the privacy of $\mathcal{D}_{\text{sanitized}}$ under a re-identification attack by an adversary which has access to $\mathcal{D}_{\text{sanitized}}$ as well as auxiliary information $\tilde{x}^{(i)} = A(x^{(i)}) \subset x^{(i)}$ for entries in $\mathcal{D}_{\text{original}}$. The access function $A$ that determines the amount and the type of auxiliary information depends on the threat model; in our experiments, we just assume that $A(x)$ randomly selects three claims from $x$.

To assess potential privacy breaches that could result from the public release of a sanitized dataset, we define $L(\tilde{x}^{(i)}, \mathcal{D}_{\text{sanitized}}) \to \hat{y}^{(i)}$ as a linking method that takes some auxiliary information $\tilde{x}^{(i)}$ and the sanitized dataset $\mathcal{D}_{\text{sanitized}}$ as inputs and produces a linked record $\hat{y}^{(i)} \in \mathcal{D}_{\text{sanitized}}$. Let $\mu(x^{(i)}, \hat{y}^{(i)})$ be a semantic distance metric quantifying the dissimilarity between the original record $x^{(i)}$ and the linked record $\hat{y}^{(i)}$. Given these components, we define our privacy metric as:

$$\begin{aligned}
&\text{privacy}(\mathcal{D}_{\text{original}}, \mathcal{D}_{\text{sanitized}}) \\
&= \mathbb{E}_{x^{(i)} \in \mathcal{D}_{\text{original}}, \tilde{x}^{(i)} \subset x^{(i)}}[\mu(x^{(i)}, L(\tilde{x}^{(i)}, \mathcal{D}_{\text{sanitized}}))].
\end{aligned} \tag{1}$$

In addition, we measure the utility of $\mathcal{D}_{\text{sanitized}}$ to explore the privacy-utility tradeoff, which we detail in §3.1.

**Linking method $L$.** We employ a sparse information retrieval technique $L_{\text{sparse}}$, specifically the BM25 retriever (Lin et al., 2021), to link auxiliary information with sanitized documents. Our approach concatenates the auxiliary information $\tilde{x}^{(i)}$ into a single text chunk, which serves as the query for searching a datastore of sanitized documents. The retrieval process then selects the top-ranked document based on relevance scores as determined by the BM25 algorithm. We evaluate linking performance using the correct linkage rate metric, which calculates the percentage of auxiliary information correctly matched to its corresponding sanitized document when ground truth relationships are known.

**Semantic distance metric** $\mu$. Upon linking auxiliary information $\tilde{x}^{(i)}$ to a sanitized document $\hat{y}^{(i)}$, we quantify the amount of information gain using a semantic distance metric $\mu_{\text{semantic}}$. This metric employs a language model to assess the semantic dissimilarity between the retrieved sanitized document $\hat{y}^{(i)}$ and its original counterpart $x^{(i)}$. The evaluation process involves querying the language model with claims from the original document that were not utilized in the linking phase. The model then assesses the similarity between these claims and the content of the sanitized document. We employ a three-point scale for this assessment: a score of 1 indicates identical information, while a score of 3 signifies that the claim is unsupported by the sanitized document. When reporting the scores, we normalize them to the range [0,1]. In this scoring scheme, a higher value of $\mu$ corresponds to a greater degree of privacy preservation, as it indicates reduced similarity between the original and sanitized documents. The specific prompt used for this evaluation can be found in Appendix G.4.

Our implementation uses LLaMA 3.1 8B (Dubey et al., 2024) to calculate the semantic distance metric $\mu$. To improve the model's consistency, we query the LLaMA model five times for each semantic distance metric evaluation and determine the final classification based on the mode of these responses. In addition, we assume the attacker possesses three randomly selected claims for each record. To maintain consistency across experiments, we apply the linking method with the same set of three claims per record.

**Baseline.** To validate our approach, we establish a baseline using established text distance metrics, defining complementary functions $L_{\text{rouge}}$ and $\mu_{\text{rouge}}$. Both functions are implemented using ROUGE-L (Lin, 2004), which is widely used in the literature as an automated metric Dou et al. (2024); Xiao et al. (2024); Frikha et al. (2024); Huang et al. (2023). Specifically, the baseline linking method $L_{\text{rouge}}$ processes auxiliary information $\tilde{x}^{(i)}$ by concatenating it into a single text chunk, following the approach described in Section 2, and identifies the sanitized document with the maximum ROUGE-L score. To compute the baseline privacy metric $\mu_{\text{rouge}}$, we calculate one minus the ROUGE-L score between the original document $x^{(i)}$ and its linked sanitized version $\hat{y}^{(i)}$. This formulation ensures that higher values indicate stronger privacy protection.

## 3 EXPERIMENTAL SETUP

### 3.1 DATASETS AND UTILITY METRICS

We use two datasets in our study: MedQA (Jin et al., 2021) and WildChat (Zhao et al., 2024). Each dataset employs distinct measures of downstream utility to assess the effectiveness of our sanitization method, which we detail below. In addition to dataset-specific evaluations, we assess the quality of sanitization across the two datasets.

#### 3.1.1 DATASETS

**MedQA dataset.** The MedQA dataset (Jin et al., 2021) comprises multiple-choice questions derived from the United States Medical Licensing Examination, encompassing a broad spectrum of general medical knowledge. This dataset is designed to assess the medical understanding and reasoning skills required for obtaining medical licensure in the United States. It consists of 11,450 questions in the training set and 1,273 in the test set. Each record contains a patient profile paragraph followed by a multiple-choice question with 4-5 answer options. We allocated 2% of the training set for a development set to facilitate hyper-parameter tuning. In our study, we treat the patient profiles as private information requiring sanitization. Given a sanitization method $S$, and for each record $x^{(i)} \in \mathcal{D}_{\text{original}}$, we generate the sanitized version of the patient profile, and task the evaluation model, LLaMA 3.1 8B (Dubey et al., 2024), with the multiple-choice question using the sanitized patient profile. We report the accuracy of this evaluator's performance as our utility metric.

**WildChat dataset.** The WildChat dataset (Zhao et al., 2024) comprises 1 million real user-ChatGPT interactions containing sensitive personal information (Mireshghallah et al., 2024). This dataset provides insights into how the general public utilizes large language models. Following the pre-processing steps outlined in Mireshghallah et al. (2024), we categorize each conversation $x^{(i)} \in \mathcal{D}_{\text{original}}$ and task the sanitization method $S$ to generate sanitized conversations. We evaluate

the distribution of categories in these generated conversations, reporting the chi-squared distance from the original distribution as a measure of utility. Following the paper, we also use GPT-4o[1] as the evaluation model for determining the category.

We scale the resulting chi-squared distance so that 1 indicates perfect distribution preservation. On the other hand, we set the value of 0 to be the chi-squared distance from the original dataset distribution to the uniform distribution of all categories. If a distribution is significantly different from the original distribution, a negative value is possible. Often, the chatbot in the user-bot interaction generate lengthy and duplicated content, especially when the user asks a question to the bot. This effect leads to atomization process containing less user supplied information. To address this complexity introduced by bot-generated content within the dataset, we implement an additional preprocessing step. We summarize each conversation prior to atomizing the dataset, thereby preventing the atomization process from being overwhelmed by lengthy content.

### 3.1.2 QUALITY OF GENERATION METRIC

We furthermore add the sanitization quality metric to our utility metric suite. Inspired by recent works (Zeng et al., 2024a; Chiang & Lee, 2023), we employ a large language model (in our case, GPT-4o) as a judge to assess the quality of sanitization outputs on a Likert scale of 1 to 5, with a specific focus on text coherence. For this metric, we utilize GPT-4o as our evaluation model. We provide our prompts used in Appendix G.

### 3.2 DATA SANITIZATION TECHNIQUES

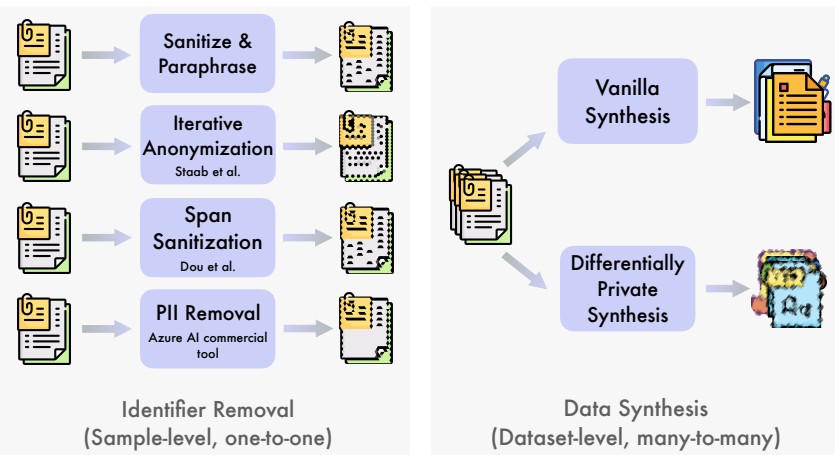

Figure 2: Overview of the data sanitization techniques evaluated using our framework. We evaluate two main categories: identifier removal methods and data synthesis methods. Identifier removal methods operate at the sample level, maintaining a one-to-one correspondence between original and sanitized records. In contrast, data synthesis methods operate at the dataset level, where each sanitized record may derive information from multiple original records.

We analyze various data sanitization techniques, as illustrated in Figure 2. Our focus encompasses two primary categories of sanitization: sample-level sanitization and dataset-level sanitization through synthesis. Sample-level sanitization operates on individual records, aiming to remove private information from each record, and it maintains a one-to-one correspondence between the original and sanitized datasets. We implement **Prompt-based Sanitization** (Staab et al., 2024), **Prompt-based Sanitization with Paraphrasing**, **Named Entity Recognition and Anonymization** (Dou et al., 2024), and **Data Sanitization via Scrubbing** in this category. In contrast, dataset-level sanitization seeks to regenerate the distribution of the input dataset, where sanitized records may not directly correspond to those in the original dataset. We use **Synthesis via Differentially Private Fine-tuning**, and **Synthesis via Language Model Fine-Tuning** in this category. We incorporate

---

[1]https://openai.com/index/hello-gpt-4o/

two additional baselines: **No Sanitization** and **Remove All Information**. Detailed description of these methods is available in Appendix C.1, and prompts used in our analysis are provided in Appendix G.

## 4 EXPERIMENTAL RESULTS

### 4.1 PRIVACY-UTILITY TRADE-OFF: IDENTIFIER REMOVAL AND DATA SYNTHESIS

Table 1 shows that both identifier removal and data synthesis methods fail to achieve perfect privacy (semantic distance of 1.0).

Identifier removal methods (Sanitize & Paraphrase, Azure AI PII tool, Span Sanitization, Iterative Anonymization) display a consistent pattern: their lexical distance values exceed their semantic distance measurements. This difference reveals that while these methods alter the surface text, they preserve the underlying semantic connections that enable inference attacks. The Azure AI tool, despite its commercial adoption, achieves only 0.11 semantic distance, indicating limited privacy protection.

Data synthesis methods reduce the gap between lexical and semantic privacy metrics compared to identifier removal approaches. However, their effectiveness varies by dataset. On MedQA, data synthesis methods achieve privacy and utility levels similar to identifier removal. On WildChat, data synthesis shows lower task utility compared to most identifier removal methods, suggesting a direct trade-off between privacy and utility.

These results demonstrate that identifier removal methods create a *false sense of privacy*, where lexical metrics report artificially higher privacy than the actual semantic information leakage. In contrast, lexical metrics more accurately reflect the privacy preservation of data synthesis methods. However, our analysis shows these methods face similar utility-privacy trade-offs comapred to identifier removal methods on MedQA and decreased utility on WildChat tasks.

### 4.2 PRIVACY-UTILITY TRADE-OFF: DATA SYNTHESIS WITH DIFFERENTIAL PRIVACY

In the previous section, we showed that data synthesis offers similar privacy-utility trade-off compared to identifier removal methods. However, this sanitization technique remains imperfect, as privacy leakage persists. To address this, researchers often integrate data synthesis with differential privacy (DP) to establish formal bounds on potential data leakage (Yue et al., 2023). The bounding of the leakage in DP is governed by the privacy budget, denoted as $\varepsilon$. A higher $\varepsilon$ value corresponds to reduced privacy. Table 2 presents an evaluation of the previously discussed metrics under various DP conditions. The row where $\varepsilon = \infty$ is equivalent to not applying differential privacy, i.e. the vanilla data synthesis row from Table 1.

Our analysis shows that applying DP improves privacy protection even with high privacy budgets like $\varepsilon = 1024$. For MedQA, the lexical privacy metric increases from $0.41$ to $0.82$, and the semantic privacy metric from $0.45$ to $0.90$. This privacy improvement creates a direct trade-off with utility. The MedQA utility decreases from $0.61$ to $0.42$, dropping below the no-private-data baseline of $0.44$.

The WildChat dataset exhibits similar utility degradation under DP. With a strict privacy budget ($\varepsilon = 3$), the utility falls below $0$, indicating that the sanitized label distribution deviates from ground truth more than a uniform distribution would. The textual coherence metric also decreases substantially from $3.28$ to $1.86$, where $1$ represents "Very Poor" quality text. We showcase an example output in Table 3. Based on this sharp decline in utility, we did not evaluate stricter privacy settings with lower $\varepsilon$ values.

Unlike the non-DP results, some $\varepsilon$ settings produce lexical privacy metrics that are lower than semantic similarity metrics. Through manual inspection, we found this occurs due to degraded text quality. These cases show minimal meaningful information leakage, with non-perfect lexical privacy scores ($< 1.0$) arising from matches in common words like articles and prepositions rather than actual private content leakage.

Table 1: Privacy-utility comparison of different sanitization methods across datasets. Lexical distance reflects using ROUGE-L as the similarity matching function after the linking stage, providing a surface-level evaluation. Sanitization methods are introduced in Section 3.2. In particular, **Span Sanitization** refers to the sanitization method proposed by Dou et al. (2024), and **Iterative Anonymization** refers to the technique proposed by Staab et al. (2024). The utility metric for the WildChat dataset is normalized to the range of [0, 1] across all sanitization methods. Our analysis shows that identifier removal methods often offer a false sense of privacy, with lexical distance metrics consistently higher than the semantic distance. The effectiveness of data synthesis methods varies by dataset–achieving comparable privacy-utility trade-offs to identifier removal methods on MedQA, but showing reduced utility on WildChat.

| | | **Privacy** ↑ | | **Utility** ↑ | |
|---|---|---|---|---|---|
| **Dataset** | **Method** | **Lexical Distance** | **Semantic Distance** | **Task Utility** | **Text Coherence** |
| | No Sanitization | $0.10_{(0.000)}$ | $0.09_{(0.004)}$ | $0.69_{(0.000)}$ | $3.79_{(0.006)}$ |
| | Remove All Info | - | - | $0.44_{(0.000)}$ | - |
| **MedQA** | Sanitize & Paraphrase | $0.72_{(0.004)}$ | $0.31_{(0.024)}$ | $0.67_{(0.012)}$ | $3.67_{(0.010)}$ |
| | Azure AI PII tool | $0.16_{(0.000)}$ | $0.11_{(0.004)}$ | $0.67_{(0.000)}$ | $3.27_{(0.012)}$ |
| | Span Sanitization | $0.61_{(0.002)}$ | $0.43_{(0.004)}$ | $0.62_{(0.012)}$ | $2.84_{(0.009)}$ |
| | Iterative Anonymization | $0.49_{(0.007)}$ | $0.39_{(0.006)}$ | $0.62_{(0.004)}$ | $3.05_{(0.019)}$ |
| | Data Synthesis | $0.41_{(0.013)}$ | $0.45_{(0.016)}$ | $0.61_{(0.007)}$ | $3.48_{(0.033)}$ |
| | No Sanitization | $0.31_{(0.000)}$ | $0.19_{(0.003)}$ | $0.96_{(0.006)}$ | $4.09_{(0.024)}$ |
| **WildChat** | Sanitize & Paraphrase | $0.66_{(0.003)}$ | $0.36_{(0.004)}$ | $0.57_{(0.014)}$ | $3.48_{(0.042)}$ |
| | Azure AI PII tool | $0.35_{(0.000)}$ | $0.22_{(0.002)}$ | $0.96_{(0.002)}$ | $3.59_{(0.008)}$ |
| | Span Sanitization | $0.47_{(0.002)}$ | $0.23_{(0.000)}$ | $0.96_{(0.003)}$ | $2.98_{(0.046)}$ |
| | Iterative Anonymization | $0.58_{(0.013)}$ | $0.41_{(0.015)}$ | $0.92_{(0.010)}$ | $3.51_{(0.027)}$ |
| | Data Synthesis | $0.86_{(0.000)}$ | $0.82_{(0.009)}$ | $0.63_{(0.020)}$ | $3.28_{(0.043)}$ |

Table 2: Privacy-utility comparison of data synthesis using differential privacy with different levels of the privacy budget $\varepsilon$, across datasets. For the WildChat dataset, the task utility is measured as the chi-squared distance between the synthesized data's label distribution and the original dataset's distribution. Values below 0 indicate that the synthesized distribution deviates substantially from the original distribution. Lower values of $\varepsilon$ provide stronger privacy guarantees. The lexical distance metric uses ROUGE-L as the similarity matching function. Differentially private sanitization methods are introduced in Section 3.2. The results demonstrate that differential privacy effectively prevents privacy leakage but yields lower utility scores compared to other methods.

| | | **Privacy** ↑ | | **Utility** ↑ | |
|---|---|---|---|---|---|
| **Dataset** | **Privacy Budget** | **Lexical Distance** | **Semantic Distance** | **Task Utility** | **Text Coherence** |
| | $\varepsilon = \infty$ | $0.41_{(0.013)}$ | $0.45_{(0.016)}$ | $0.61_{(0.007)}$ | $3.48_{(0.033)}$ |
| **MedQA** | $\varepsilon = 1024$ | $0.82_{(0.002)}$ | $0.90_{(0.004)}$ | $0.42_{(0.014)}$ | $2.23_{(0.019)}$ |
| | $\varepsilon = 64$ | $0.83_{(0.003)}$ | $0.91_{(0.003)}$ | $0.42_{(0.008)}$ | $2.14_{(0.026)}$ |
| | $\varepsilon = 3$ | $0.84_{(0.001)}$ | $0.91_{(0.003)}$ | $0.41_{(0.006)}$ | $2.04_{(0.009)}$ |
| | $\varepsilon = \infty$ | $0.86_{(0.000)}$ | $0.82_{(0.009)}$ | $0.63_{(0.020)}$ | $3.28_{(0.043)}$ |
| **WildChat** | $\varepsilon = 1024$ | $0.89_{(0.000)}$ | $0.88_{(0.008)}$ | $0.45_{(0.051)}$ | $1.86_{(0.039)}$ |
| | $\varepsilon = 64$ | $0.89_{(0.000)}$ | $0.88_{(0.002)}$ | $0.06_{(0.035)}$ | $1.86_{(0.015)}$ |
| | $\varepsilon = 3$ | $0.89_{(0.000)}$ | $0.89_{(0.003)}$ | $-0.46_{(0.102)}$ | $1.63_{(0.032)}$ |

## 4.3 ANALYSIS: CHANGING THE AVAILABLE AUXILIARY INFORMATION

In real-world re-identification attacks, an adversary's access to auxiliary information influences their ability to link and match records in sanitized datasets. For example, the first three claims in a MedQA record tend to contain the patient's age and sex information, whereas later claims tend to not have these information. Our previous experiments randomly selected three claims from each record as the

Table 3: A medical record generated by the DP sanitization method with $\varepsilon = 3$. We note that the record suffers from semantic inconsistencies, including contradictory statements about the patient's health status and redundant physical examination mentions. These artifacts are typical of DP-generated text where coherence is compromised to maintain privacy guarantees.

---

A Sample Medical Record Generated by the DP Sanitization Method with $\varepsilon = 3$:

---

A 21-year-old man presents to his family physician for evaluation...On physical examination, he is in good general health and **his physical examination reveals no abnormalities**. His pulse is 116/min. His temperature is 37.7°C (100.4°F), blood pressure is 103/73 mm Hg, and body weight is 62 kg (139 lb). **Physical examination shows generalized tenderness throughout the back and extremities**, along with an intermittent, tender warmth on the neck and forehead ...**Examination of his abdomen reveals a 4-mm-long papillary mass** ...

---

Table 4: Comparison of successful linkage rates for various data sanitization methods across datasets, assuming access to different auxiliary information (claims) for performing matching and retrieval in re-identification attempts. Sanitization methods are introduced in Section 3.2. In the MedQA dataset, the first three claims often contain a fixed set of information, such as the age, sex, and chief complaint of the patient; whereas the last three claims don't have such an information and are filled with arbitrary facts such as lab results. The high variance in these rates highlights the impact that available auxiliary side-information has on potential data leakage.

| Dataset | Method | First Three Claims | Random Three Claims | Last Three Claims |
|---|---|---|---|---|
| **MedQA** | No Sanitization | $0.99_{(0.000)}$ | $0.99_{(0.000)}$ | $0.98_{(0.000)}$ |
| | Sanitize & Paraphrase | $0.47_{(0.053)}$ | $0.73_{(0.028)}$ | $0.81_{(0.001)}$ |
| | Azure AI PII tool | $0.95_{(0.000)}$ | $0.99_{(0.000)}$ | $0.98_{(0.000)}$ |
| | Span Sanitization | $0.75_{(0.012)}$ | $0.75_{(0.002)}$ | $0.73_{(0.007)}$ |
| | Iterative Anonymization | $0.71_{(0.018)}$ | $0.79_{(0.006)}$ | $0.82_{(0.006)}$ |
| **WildChat** | No Sanitization | $0.88_{(0.000)}$ | $0.89_{(0.000)}$ | $0.85_{(0.000)}$ |
| | Sanitize & Paraphrase | $0.71_{(0.005)}$ | $0.74_{(0.006)}$ | $0.71_{(0.008)}$ |
| | Azure AI PII tool | $0.87_{(0.000)}$ | $0.87_{(0.000)}$ | $0.83_{(0.000)}$ |
| | Span Sanitization | $0.87_{(0.003)}$ | $0.89_{(0.001)}$ | $0.84_{(0.003)}$ |
| | Iterative Anonymization | $0.63_{(0.014)}$ | $0.71_{(0.020)}$ | $0.71_{(0.010)}$ |

adversary's accessible information. To assess the impact of this choice, we conducted experiments using both randomly selected claims and the first three claims.

Table 4 presents the results of these experiments, focusing on the correct linkage rate (defined in §2) for sample-level, identifier removal methods. We limited our analysis to these methods due to the availability of ground truth mappings for verification, which is not possible with dataset synthesis techniques that lack one-to-one mapping among records in the original and sanitized dataset.

In MedQA, there are structured patterns with consistent medical attributes – 89% of records contained patient age, 81% included specific symptoms, and 63% contained medical history information. This structured nature made the atomization process more systematic – we could reliably separate claims about symptoms, medical history, and demographics in an orderly fashion. However, this revealed a key privacy challenge: even after sanitization, these medical attributes are still related to each other, making re-identification easier through these linked attributes. This was particularly problematic due to the sparsity of specific age-symptom-history combinations in medical data – unique combinations of these attributes could often identify a single patient even when individually sanitized. On the other hand, data records in the WildChat dataset does not have such a strong coupling among atomized claims, as the user might change the conversation topic and that they are more general.

Results demonstrate that the type of auxiliary information that the adversary have access to is important to the linking stage, and this leads to insights into the sanitization ability of various methods. For the MedQA dataset, methods relying on LLMs, such as sanitize & paraphrase and the approach

proposed by Staab et al. (2024), exhibit the highest variance between the first three claims and the last three. claims. We hypothesize that this phenomenon may be attributed to LLMs are better at sanitizing a certain types of information, such as the age of the patient that is more prevalent in the earlier claims, resulting in uneven preservation of information across different sections of the text.

### 4.4 HUMAN EVALUATION OF THE SEMANTIC DISTANCE METRIC

To validate our language model's performance in measuring the semantic distance metric $\mu$ defined in Section 2, we conducted a controlled human evaluation study. Three authors independently annotated 580 identical claims, working without access to any model-generated outputs to prevent bias. The evaluation yielded strong inter-annotator reliability, with a Fleiss' kappa coefficient of 0.87. When comparing model performance to human judgments, we found LLaMA 3 8B achieved a Spearman correlation coefficient of 0.95 with the mode of human annotations. This performance approaches that of GPT-4, which achieved a coefficient of 0.97. For comparison, the ROUGE algorithm showed weaker alignment with human judgments, reaching a Spearman coefficient of 0.81.

Table 5: Inter-rater agreement and model correlations for semantic similarity inference task.

| Metric/Model | Measure | Value |
| --- | --- | --- |
| Human Agreement | Fleiss' Kappa | 0.875 |
| LLaMA 3 8B | Spearman Correlation | 0.919 |
| GPT-4o | Spearman Correlation | 0.946 |
| ROUGE-L recall | Spearman Correlation | -0.806 |

## 5 CONCLUSION

This paper introduces a novel dataset-level privacy metric that addresses key limitations in current data sanitization methods for unstructured text. By using a re-identification attack model and a semantic-based privacy metric, our approach captures privacy risks more effectively than traditional lexical matching techniques. Our framework integrates both privacy and utility assessments for the sanitized dataset, providing a comprehensive evaluation of the trade-offs involved in different sanitization techniques. Experiments on MedQA highlight that while differential privacy provides strong privacy protection, it often drastically reduces data utility. Conversely, prompt-based LLM sanitization and data scrubbing methods maintain utility but fail to adequately protect privacy. Fine-tuning offers similar privacy-utility trade-off compared to identifier removal methods on MedQA dataset while suffers from low utility on the WildChat dataset. Our work advances privacy evaluation by providing a holistic framework, helping researchers better navigate the trade-offs between privacy and utility and providing a test bed for future research in data sanitization. Our experiments reveal that existing sanitization methods often create *a false sense of privacy* by implementing surface-level text modifications without addressing deeper semantic vulnerabilities. The results highlight the urgent need for new privacy protection methods that specifically target semantic information leakage while preserving utility.

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

## A  RELATED WORK

**Privacy evaluations of dataset disclosure.** Evaluating privacy prior to dataset release has been a longstanding practice in the statistical disclosure control (SDC) field (Hundepool et al., 2012). This practice spans various domains, including legal, technical, and medical domains (Bellovin et al., 2019; Garfinkel, 2015; Giuffrè & Shung, 2023). Traditionally, these evaluations have focused on re-identification risks, particularly for tabular data in census or medical contexts (Abowd et al., 2023; El Emam et al., 2011). While there have been attempts to create text anonymization benchmarks (Pilán et al., 2022), these primarily concentrate on span detection and anonymization rather than re-identification and focus on scrubbing methods rather than data synthesis, contrary to our work. Recent work in the security literature has begun to challenge the perceived safety of synthetic data (Stadler et al., 2022; Yale et al., 2019; Annamalai et al., 2024), raising concerns about the privacy guarantees of synthetic data. However, these investigations primarily focused on simple, low-dimensional tabular or image data and have not extended to unstructured text, leaving a critical gap.

**Data sanitization through removal of identifiers.** Traditional approaches to data sanitization have centered on the detection and removal of Personally Identifiable Information (PII) (Mendels et al., 2018; Montani et al., 2022) relying on named entity recognition (NER) systems and masking. Recently, LLMs have been employed for this task: Staab et al. (2024) developed an iterative prompting method using GPT-4 to achieve implicit attribute removal, moving beyond simple token replacement. Similarly, Dou et al. (2024) proposed a two-step approach, combining a self-disclosure detection model with an abstraction technique to reduce privacy risks in text data. Other sanitization methods involve identifying sensitive words prompting an LLM (Zhou et al., 2024). Morris et al. (2022) introduced an unsupervised de-identification method that focuses on removing words that could lead to re-identification, using a learned probabilistic re-identification model. Similar to ours, their approach does not rely on specific rule lists of named entities but instead learns from aligned descriptive text and profile information. However, their method requires a dataset of aligned text and profiles, which may not always be available in real-world scenarios. These approaches mostly sanitize the dataset by abstracting or removing detected keywords to minimize re-identification, and are susceptible to our proposed semantic re-identification attack.

**Data sanitization through synthesis.** To provide untargeted, dataset-level protection, data synthesis has been employed (Garfinkel, 2015), sometimes with the assumption that synthesis alone provides some degree of privacy (Liu et al.). For a more principled way of providing formal privacy guarantees, differentially private (DP) data synthesis techniques have been developed, including differentially private generative adversarial network for tabular data synthesis (Xie et al., 2018; Torkzadehmahani et al., 2019). For textual data, prior work proposed and benchmarked differentially private VAE, BART, and autoencoder with embedding rewards (Weggenmann et al., 2022; Igamberdiev & Habernal, 2023; Bo et al., 2021; Igamberdiev et al., 2022), and Yue et al. (2023); Mattern et al. (2022); Mireshghallah et al. (2022); Kurakin et al. (2023) introduced differentially private fine-tuning approaches for large language models to generate synthetic text. Pang et al. (2024) and Morris et al. (2024) has shown that DP sanitized record can still be linked to the original records, but we further show that DP methods hinder utility as well. Ramesh et al. (2024) explores the privacy-utility trade-off and fairness issues of DP methods on simple classification tasks, using canary evaluation and PII detection to evaluate privacy preservation. In contrast, we provide a more general method-agnostic and task-agnostic framework for evaluating *semantic* privacy under utility constraints.

## B  LIMITATIONS

Our study is not exhaustive, and particularly it does not encompass all possible privatization techniques, such as model unlearning techniques where it is not readily applicable to the data sanitization setting. Additionally, our analysis was primarily confined to datasets within the medical and conversational domains, which limits the generalizability of our findings. Future research should focus on evaluating the method's applicability across diverse datasets and domains to establish its broader relevance and robustness.

A key challenge in our work is that the definition of privacy and what constitutes a privacy leak is often blurry and context-dependent. Privacy is fundamentally based on outcomes and how people feel about information disclosure, rather than purely objective measures or monetary harm. Our metric measures semantic similarity, which may be more relevant for some types of information (e.g., medical conditions) but less meaningful for others (e.g., social security numbers). This limitation is particularly relevant when comparing our method to techniques specifically designed for PII removal. Furthermore, there is an inherent ambiguity in distinguishing between information learned from the sanitized dataset and information that can be inferred from the auxiliary data. For example, if the auxiliary data suggests that someone is going through mental status examination, one might infer a high probability of mental disease without accessing the sanitized data. Disentangling these sources of information is challenging and not fully addressed in our current framework.

Our work does not pass judgment on whether or not inferences from the auxiliary data are privacy violations as some might be necessary for maintaining downstream utility. Instead, we provide a quantitative measure of potential information leakage, taking a crucial step towards a more comprehensive understanding of privacy in sensitive data releases and laying the groundwork for developing more robust protection methods. Ideally, a more desirable solution would be a *contextual* privacy metric, which can take into account (i) which information is more privacy-relevant and (ii) which information is private in the context that the textual information is being shared. These are challenging questions that we believe are beyond the scope of this paper. Nevertheless, they represent exciting research directions to pursue, particularly given recent advances in LLMs.

## C   IMPLEMENTATION DETAILS

We use two datasets in our study: MedQA (Jin et al., 2021) and WildChat (Zhao et al., 2024). Each dataset employs distinct measures of downstream utility to assess the effectiveness of our sanitization method, which we detail below. In addition to dataset-specific evaluations, we assess the quality of sanitization across the two datasets.

### C.1   DATA SANITIZATION TECHNIQUES

We use our metrics to evaluate two categories of sanitization methods, as illustrated in Figure 2. Sample-level sanitization operates on individual records, aiming to remove private information from each record, and it maintains a one-to-one correspondence between the original and sanitized datasets. In contrast, dataset-level sanitization seeks to create a new dataset that preserves the the textual patterns and linguistic characteristics of the input dataset, where sanitized records may not directly correspond to those in the original dataset. Detailed prompts used in our analysis are provided in Appendix G.

**Iterative anonymization (Staab et al., 2024).**   This approach utilizes LLMs to remove sensitive information through iterative prompting. We implement the sanitization pipeline proposed by Staab et al. (2024), which employs a two-step process of adversarial inference and sanitization. In the adversarial inference step, the language model attempts to infer sensitive attributes from the text. Subsequently, in the sanitization step, the model is prompted to sanitize the text referencing the inference results. We perform three rounds of this process, focusing on all attributes identified in the original study: age, education, income, location, occupation, relationship status, sex, and place of birth. For this sanitization method, we employ GPT-4o as our LLM.

**Sanitize and paraphrase.**   drawing insights from Zeng et al. (2024b), who explored record rewriting, we implement a sequential privacy protection approach. we first apply the sanitization prompt from Staab et al. (2024) without attribute inference, then use GPT-4o to paraphrase the sanitized text, potentially enhancing privacy protection.

**Span sanitization (Dou et al., 2024).**   we evaluate the self-disclosure detection model developed by Dou et al. (2024). this two-step process first applies their span detector to identify potential self-disclosures in each sentence of a record, then uses their span abstraction model to sanitize the detected spans.

**Azure AI PII tool.**   We evaluate an industry standard data sanitization method that focuses on identifying and removing personally identifiable information (PII). This approach utilizes the Azure AI Language PII detection service[2] to identify and redact PII from the dataset with the "*" character.

**Data synthesis via differentially private fine-tuning.**   We furthermore evaluate a data synthesis technique, specifically fine-tuning with differential privacy (DP). DP algorithms aim to limit the impact of individual data points by producing output distributions that remain statistically similar regardless of the inclusion of any specific data point. We adopt the method described by Yue et al. (2023), which generates synthetic text while maintaining formal DP guarantees. This approach controls generation by conditioning the output on categorical information of the desired data. Prior to fine-tuning a generative model, the method preprocesses data records by prepending a "control code", a categorical label, to each record. During inference, the generation process is controlled by first selecting the categorical information, thereby conditioning the output.

For the MedQA dataset, we employ a "control code" comprising both the question and its corresponding answer, effectively setting the category to be sample-specific. Specifically, we prepend a text snippet in the format "Question: [question text] |Answer: [answer text]" to each record $x^{(i)}$. During the generation of sanitized records, we provide this same text snippet and ask the model to generate the corresponding record, treating the generated record as the sanitized information.

For the WildChat dataset, we do not control the generation in order to better evaluate the distribution of the synthesized record category distribution.

In our experiments, we apply this method to our datasets with privacy budget values of $\varepsilon \in \{3, 8, 16, 64, 512, 1024\}$ that are commonly used in the differential privacy literature.

**Data synthesis via language model fine-tuning.**   We implement a data processing pipeline following the approach outlined in "Synthesis via Differentially Private Fine-tuning." The implementation uses the control code mechanism described above, with standard fine-tuning parameters: an unbounded privacy budget ($\epsilon = \infty$) and disabled gradient clipping.

**Sanitization baselines.**   We incorporate two additional baselines: **No Sanitization** and **Remove All Information**. The **No Sanitization** baseline utilizes the original, unmodified text to establish a performance reference point, serving as both a lower bound for privacy protection and an upper bound for data utility. Conversely, the **Remove All Information** baseline, evaluated on MedQA, sanitize the text by removing the provided record, measuring the underlying knowledge and inherent biases of the language model.

# D   ADDITIONAL EXPERIMENTS

## D.1   ABLATION ON LINKER

We conduct experiments on different linker designs, focusing on two key aspects: comparing retrieval methods and evaluating strategies to construct retriever queries with the auxiliary information. Our analysis contrasts BM25, a lexical retriever, with GritLM (Muennighoff et al., 2024), a semantic retriever, while also examining different approaches to construct the query for the retriever.

**Varying retriever.**   Our baseline implementation uses BM25 (Lin et al., 2021), a sparse retriever that matches auxiliary information to sanitized documents using term frequency-inverse document frequency (TF-IDF) weighting. This approach computes relevance scores between query terms and documents to identify the most relevant matches. We compare this against GritLM (Muennighoff et al., 2024), a state-of-the-art semantic retriever that embeds both records and candidates in a high-dimensional vector space and retrieves nearest neighbors based on semantic similarity.

**Varying query construction from auxiliary information.**   we evaluate two approaches to construct queries from auxiliary information. our primary method merges all auxiliary information into

---

[2]`https://learn.microsoft.com/en-us/azure/ai-services/language-service/personally-identifiable-information/overview`

a single query. the alternative approach treats each piece of auxiliary information $\tilde{x}^{(i)}$ as an independent query against the database of atomized sanitized documents. in this second approach, the retriever identifies similar claims from the sanitized dataset $\mathcal{D}_{\text{sanitized}}$ for each auxiliary information claim. the final document selection uses majority voting, selecting the document that most frequently matches across all auxiliary information claims.

Table 6: Comparison of successful linkage rates for various linker designs. We bold the highest performing linker across various sanitization methods. We report the standard deviation as a result of three separate seeds. Sanitization methods are introduced in Section 3.2. In particular, **Span Sanitization** refers to sanitizaiton method proposed by (Dou et al., 2024), and **Iterative Anonymization** refers to the technique proposed by (Staab et al., 2024). We note that our choice of linker outperforms other linker designs on most of the sanitization methods. .

| | Sanitization Method | BM25 Matching with Single Query (ours) | BM25 Matching with Majority Voting | Grit Matching with Single Query | Grit Matching with Majority Voting |
|---|---|---|---|---|---|
| **MedQA** | No Sanitization | **0.99**$_{(0.000)}$ | 0.97$_{(0.000)}$ | 0.74$_{(0.000)}$ | **0.99**$_{(0.001)}$ |
| | Sanitize & Paraphrase | 0.73$_{(0.028)}$ | 0.56$_{(0.014)}$ | 0.69$_{(0.005)}$ | **0.78**$_{(0.013)}$ |
| | Azure AI PII tool | **0.99**$_{(0.000)}$ | 0.89$_{(0.003)}$ | 0.75$_{(0.000)}$ | 0.91$_{(0.001)}$ |
| | Span Sanitization | **0.75**$_{(0.002)}$ | 0.52$_{(0.013)}$ | 0.67$_{(0.006)}$ | 0.66$_{(0.011)}$ |
| | Iterative Anonymization | **0.79**$_{(0.006)}$ | 0.63$_{(0.003)}$ | 0.61$_{(0.011)}$ | 0.69$_{(0.010)}$ |
| **WildChat** | No Sanitization | 0.89$_{(0.000)}$ | 0.97$_{(0.000)}$ | 0.88$_{(0.000)}$ | **0.98**$_{(0.000)}$ |
| | Sanitize & Paraphrase | 0.74$_{(0.006)}$ | 0.67$_{(0.019)}$ | **0.81**$_{(0.005)}$ | 0.77$_{(0.011)}$ |
| | Azure AI PII tool | **0.87**$_{(0.000)}$ | 0.86$_{(0.007)}$ | 0.86$_{(0.000)}$ | **0.87**$_{(0.006)}$ |
| | Span Sanitization | **0.89**$_{(0.001)}$ | 0.86$_{(0.005)}$ | 0.87$_{(0.003)}$ | 0.88$_{(0.018)}$ |
| | Iterative Anonymization | 0.71$_{(0.020)}$ | 0.63$_{(0.008)}$ | **0.75**$_{(0.009)}$ | 0.71$_{(0.014)}$ |

Results in Table 6 show that BM25 with merged atom queries performs better than other linkers on most sanitization methods on the MedQA dataset. This effectiveness stems from two factors: MedQA's sparse nature and the preservation of unique medical terms during sanitization, which together enable strong performance in sparse retrieval. The majority voting approach with BM25 shows reduced performance, likely due to the uneven distribution of terms across atomized documents. In contrast, majority voting enhances linking accuracy when used with the semantic GritLM retriever. We attribute this to the semantic retriever's improved performance when matching single pieces of information, particularly in sparse datasets like MedQA.

The WildChat dataset presents different patterns. As a dataset of user-chatbot interactions, it exhibits lower vocabulary sparsity compared to MedQA. This characteristic enables semantic linkers to achieve comparable performance to sparse retrievers. The merged query approach performs at least as well as majority voting across most sanitization techniques, except for the no-sanitization condition. We attribute this to WildChat documents typically containing unified themes, where comprehensive information provides better context for matching. This contrasts with MedQA, where individual pieces of auxiliary information may have weaker interconnections.

### D.2 ANALYSIS OF CATEGORIES OF DETECTED PRIVACY LEAKAGE

We investigate the types of privacy leakage associated with each sanitization method. We adapt privacy categories from Dou et al. (2024). For the MedQA dataset, which primarily contains health-related content, we created specialized subcategories based on the History and Physical Examination guidelines from Goldberg.

To categorize privacy leakage of various sanitization methods, we used GPT-4 to analyze each claim in the original dataset. We considered a privacy leak to occur when a sanitized document supported a claim with a privacy score of 2 or higher, as defined in Section 2. We then tracked the total number of leakage across all categories for each sanitization method.

Table 7 presents the list of categories that we consider in this work, while Figure 3 shows the leakage for each sanitization method. Our analysis reveals distinct patterns across datasets. The data synthesis approach showed varying effectiveness: it removed half the sensitive attributes in MedQA and nearly all in WildChat, reflecting differences in the underlying attack models. Differential privacy sanitization methods effectively removed most sensitive information leakage, validating the

Table 7: Sensitive information categories for classifying privacy leakage types in the dataset.

| Category | Description |
| --- | --- |
| Age | Any mention of a person's age, e.g., "23-year-old" |
| Gender | References to gender identity, e.g., "woman," "non-binary person" |
| Sexual_Orientation | Mentions of sexual orientation, e.g., "gay couple" |
| Race_Nationality | References to race, ethnicity, or nationality |
| Spouse | Mentions of a person's wife, husband, or spouse |
| Partner | References to a person's girlfriend, boyfriend, or partner |
| Relationship_Status | Mentions of marital status, being in a romantic relationship, or being single |
| Family | References to family members or family structures |
| Health (Only used in Wild-Chat) | Includes a wide range of health-related information, from specific diseases or conditions to medications, medical tests, or treatments |
| Mental_Health (Only used in WildChat) | Includes a broad range of emotional states and mental health conditions, from feelings of sadness or anxiety to specific diagnoses |
| Location | Captures specific geographical details about where a person lives or is located. Includes precise locations such as addresses, cities, countries, or distinctive landmarks |
| Appearance | Physical descriptions of individuals, e.g., "He is 6'2" |
| Pet | Information about a person's pets or animals |
| Occupation | References to a person's job or profession |
| Education | Information about a person's educational background or current studies |
| Finance | Any details about financial situations or status, not necessarily exact amounts |
| **MedQA Specific** | |
| Chief_Concern | The primary reason for a medical visit or the main health issue |
| History_of_Present_Illness | Detailed account of the development of the current health problem |
| Past_Medical_History | Previous illnesses, surgeries, or significant health events |
| Medications | Current or past medications, including dosages and frequencies |
| Allergies_Reactions | Any known allergies or adverse reactions to medications or substances |
| Social_History | Information about lifestyle, habits, occupation, and living situation that may impact health |
| Family_History | Health information about immediate family members |
| Review_of_Systems | Systematic review of body systems for additional symptoms |
| Physical_Exam | Findings from a physical examination |
| Diagnostic_Results | Results from laboratory tests (blood, urine, etc.), radiologic studies (X-rays, CT scans, MRIs, etc.), and other diagnostic procedures (e.g., EKG interpretations) |

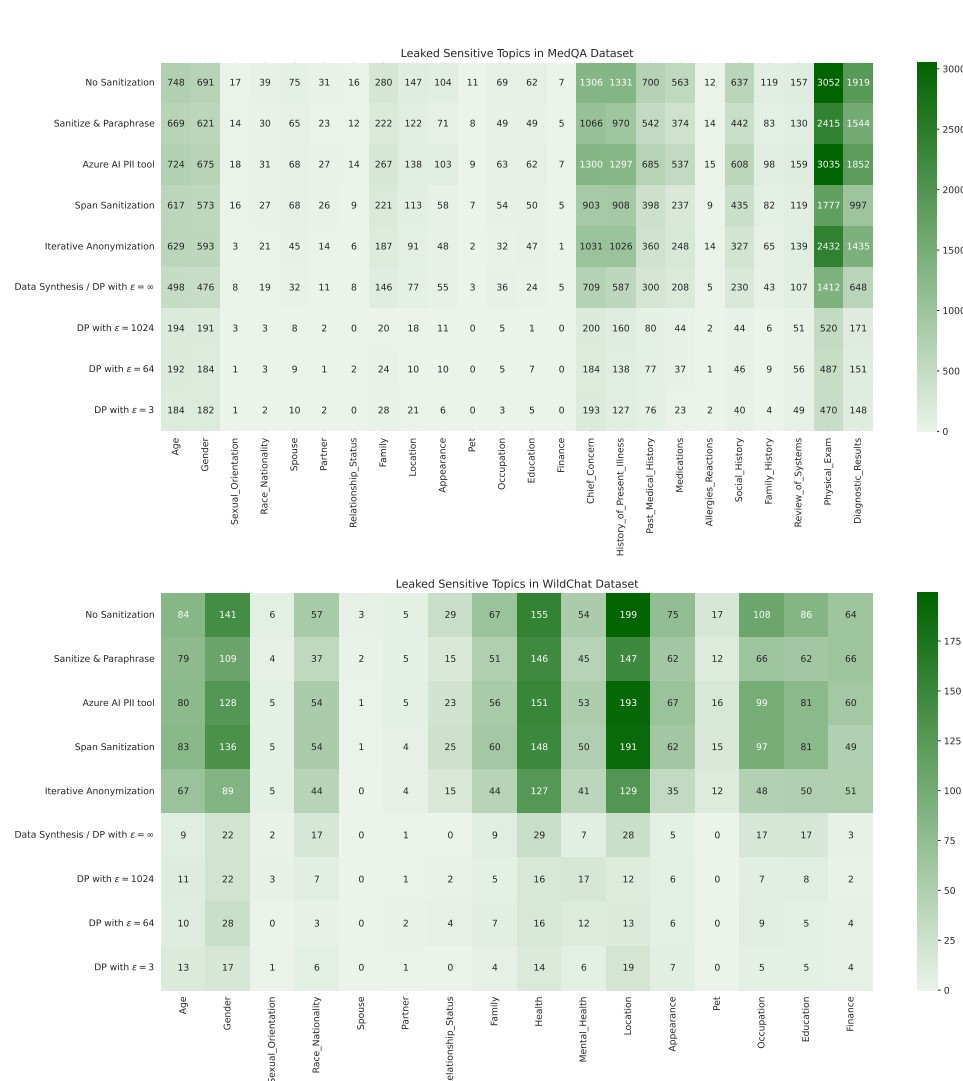

Figure 3: Distribution of leaked sensitive categories for each of the sanitization methods (Section 3.2) on the MedQA and WildChat dataset. **Span Sanitization** refers to sanitizaiton method proposed by (Dou et al., 2024), and **Iterative Anonymization** refers to the technique proposed by (Staab et al., 2024).

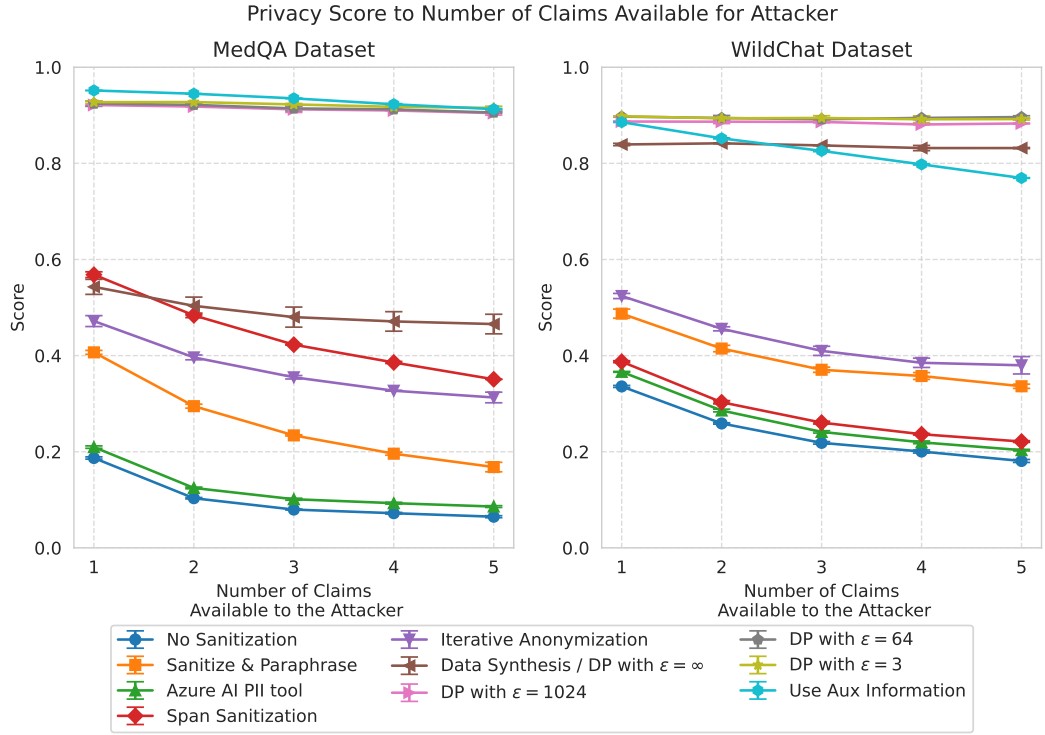

Figure 4: Privacy scores to the number of claims available to the attacker across different sanitization methods (Section 3.2). Sanitization methods are introduced in Section 3.2. In particular, **Span Sanitization** refers to sanitizaiton method proposed by (Dou et al., 2024), and **Iterative Anonymization** refers to the technique proposed by (Staab et al., 2024). The **Use Aux Information** row quantifies the information overlap between auxiliary information provided to attackers and the remaining document content.

privacy protection capabilities of differential privacy methods. On the other hand, identifier removal methods, such as Advanced Anonymizer (Staab et al., 2024) or Span Sanitizer (Dou et al., 2024), performed well on common sensitive attributes like age and gender but showed limitations with specialized medical data. We attribute this to the method's dependency on predefined category lists for sanitization, which requires careful curation for each dataset. In this case, The findings show that our privacy metric can help sanitization method designers identify overlooked categories when privacy scores indicate inadequate protection.

### D.3 ANALYSIS ON INFORMATION AVAILABLE TO THE ATTACKER

We examine how our method's effectiveness varies with both the quantity of information available to the attacker and the information overlap between auxiliary information and the rest of the record.

We first explore privacy score degradation as attackers gain access to more information. Instead of providing three random claims from a record, we provide the last $k$ claims to the attacker, where $k \in \{1, 2, 3, 4, 5\}$, and measure the resulting privacy score.

Then, we investigate the amount of information overlap between claims during atomization. Claims often share partial information when they describe different attributes of the same object. This overlap can provide attackers with additional information beyond the explicitly provided data during the evaluation. To measure this overlap, we apply the semantic distance metric defined in Section G.4, treating the provided auxiliary information as the sanitized document while maintaining the standard evaluation procedure. In this context, a higher privacy score indicates reduced information overlap between the auxiliary information and the rest of the document.

Figure 4 presents both the privacy metric degradation and the information overlap results, with overlap reported as **Use Aux Information**. Methods without theoretical guarantees show decreased privacy as attacker information increases, with the steepest decline occurring when adding claims to case of the last one or two claims. This decline slows with additional claims, supporting our choice to use three claims for sanitization method evaluation. In contrast, DP methods maintain consistent performance regardless of the number of claims available to attackers, demonstrating their robust privacy protection. The information overlap analysis reveals modest overlap levels, with MedQA dataset showing overlap above 0.9 and WildChat at 0.78. The overlap decreases linearly as the amount of provided information increases.

## D.4 COMPARISON TO ALTERNATIVE METRICS

Table 8: Comparison of our proposed metric to three other metrics: **MAUVE**, **Embedding**, and **PII Existence**. Sanitization methods are introduced in Section 3.2. In particular, **Span Sanitization** refers to sanitizaiton method proposed by (Dou et al., 2024), and **Iterative Anonymization** refers to the technique proposed by (Staab et al., 2024).

| | Sanitization Method | Mauve | Embedding | PII Existence | Lexical Distance | Semantic Distance (Ours) |
|---|---|---|---|---|---|---|
| | No Sanitization | $0.00_{(0.000)}$ | $0.00_{(0.000)}$ | $0.00_{(0.000)}$ | $0.10_{(0.000)}$ | $0.09_{(0.004)}$ |
| **MedQA** | Sanitize & Paraphrase | $0.93_{(0.007)}$ | $0.33_{(0.009)}$ | $0.78_{(0.008)}$ | $0.72_{(0.004)}$ | $0.31_{(0.024)}$ |
| | Azure AI PII tool | $0.51_{(0.000)}$ | $0.09_{(0.001)}$ | $0.99_{(0.000)}$ | $0.16_{(0.000)}$ | $0.11_{(0.004)}$ |
| | Span Sanitization | $0.79_{(0.021)}$ | $0.33_{(0.004)}$ | $0.62_{(0.008)}$ | $0.61_{(0.002)}$ | $0.43_{(0.004)}$ |
| | Iterative Anonymization | $0.74_{(0.036)}$ | $0.34_{(0.001)}$ | $0.78_{(0.006)}$ | $0.49_{(0.007)}$ | $0.39_{(0.006)}$ |
| | Data Synthesis / $\varepsilon = \infty$ | $0.01_{(0.005)}$ | $0.26_{(0.013)}$ | $0.15_{(0.018)}$ | $0.41_{(0.013)}$ | $0.45_{(0.016)}$ |
| | DP with $\varepsilon = 1024$ | $0.17_{(0.018)}$ | $0.55_{(0.002)}$ | $0.90_{(0.003)}$ | $0.82_{(0.003)}$ | $0.90_{(0.004)}$ |
| | DP with $\varepsilon = 64$ | $0.25_{(0.040)}$ | $0.55_{(0.002)}$ | $0.89_{(0.005)}$ | $0.83_{(0.003)}$ | $0.91_{(0.003)}$ |
| | DP with $\varepsilon = 3$ | $0.38_{(0.017)}$ | $0.56_{(0.003)}$ | $0.89_{(0.014)}$ | $0.84_{(0.001)}$ | $0.91_{(0.003)}$ |
| | No Sanitization | $0.00_{(0.000)}$ | $0.05_{(0.000)}$ | $0.00_{(0.000)}$ | $0.31_{(0.000)}$ | $0.19_{(0.003)}$ |
| **WildChat** | Sanitize & Paraphrase | $0.80_{(0.011)}$ | $0.40_{(0.004)}$ | $0.41_{(0.007)}$ | $0.66_{(0.003)}$ | $0.36_{(0.004)}$ |
| | Azure AI PII tool | $0.26_{(0.000)}$ | $0.29_{(0.001)}$ | $0.69_{(0.000)}$ | $0.35_{(0.000)}$ | $0.22_{(0.002)}$ |
| | Span Sanitization | $0.72_{(0.005)}$ | $0.28_{(0.000)}$ | $0.10_{(0.001)}$ | $0.47_{(0.002)}$ | $0.23_{(0.000)}$ |
| | Iterative Anonymization | $0.81_{(0.003)}$ | $0.43_{(0.011)}$ | $0.34_{(0.016)}$ | $0.58_{(0.013)}$ | $0.41_{(0.015)}$ |
| | Data Synthesis / $\varepsilon = \infty$ | $0.95_{(0.011)}$ | $0.63_{(0.001)}$ | $0.48_{(0.016)}$ | $0.86_{(0.000)}$ | $0.82_{(0.009)}$ |
| | DP with $\varepsilon = 1024$ | $0.87_{(0.020)}$ | $0.66_{(0.002)}$ | $0.50_{(0.022)}$ | $0.89_{(0.000)}$ | $0.88_{(0.008)}$ |
| | DP with $\varepsilon = 64$ | $0.89_{(0.008)}$ | $0.66_{(0.004)}$ | $0.52_{(0.016)}$ | $0.89_{(0.000)}$ | $0.88_{(0.002)}$ |
| | DP with $\varepsilon = 3$ | $0.88_{(0.015)}$ | $0.68_{(0.002)}$ | $0.58_{(0.014)}$ | $0.89_{(0.000)}$ | $0.89_{(0.003)}$ |

We evaluate our metrics against other established approaches in measuring privacy preservation, including distributional, embedding-based, and identifier-based metrics.

**MAUVE.** We use MAUVE (Pillutla et al., 2021) to measure the difference between original and sanitized texts using divergence frontiers. This metric does not utilize auxiliary information linking, and instead directly measuring differences between the original and sanitized datasets.

**Embedding.** We use the all-MiniLM-L6-v2 model (Wang et al., 2020) to compute embedding distances between linked original and sanitized documents. We first embed each claim from both original and sanitized documents. Then, for claims not used for linking in the original document, we compute dot products of the selected claim embedding with all sanitized claims and select the maximum score. The final metric represents the mean score across all original document claims.

**PII existence.** This baseline metric examines personally identifiable information (PII) detected by Azure AI, excluding information used for document linking. We calculate the match rate between original and sanitized documents for each PII instance.

**Lexical and semantic distance.** We include these metrics from Section 4.1 as reference points for our comparison.

The results are shown in Table 8, revealing limitations in existing metrics. MAUVE is inadequate for privacy preservation measurement. For example, in the MedQA dataset, MAUVE reports that Data Synthesis sanitization leaks all information, and it suggests that the PII sanitization is more private

compared to Data Synthesis method, achieving a score of 0.5. However, upon manual inspection, it is clear that PII sanitization leaks more information than Data Synthesis. This discrepancy stems from MAUVE's focus on token distribution at the dataset level, ignoring individual record privacy. The embedding metric, while operating at the record level, is harder to interpret when compared to our semantic distance metric. The maximum score of 0.68 lacks clear privacy implications. PII Existence metrics suggest strong privacy preservation for the PII removal method, particularly in the MedQA dataset. However, our analysis reveals that PII sanitization provides little privacy protection, contrary to what this metric suggest.

### D.5 DISTRIBUTION OF PRIVACY SCORES FOR SANITIZATION METHODS

We report the privacy score distribution of the existing data sanitization methods, and the results are shown in Figure 5. We observe that identifier removal sanitization methods demonstrate significant vulnerabilities, with multiple records exhibiting complete information leakage, indicating poor worst-case privacy protection. Most methods in this category show a concentration of privacy scores at 1.0, representing maximum privacy. Manual inspection of these high-scoring records indicates that this privacy preservation stems from linker failures, where the provided auxiliary information fails to locate the target document.

Differentially private documents consistently demonstrate strong privacy preservation with minimal information leakage. However, a small subset of these documents shows unexpectedly low privacy scores. Manual analysis reveals that these anomalies result from language model hallucinations, which incorrectly indicate privacy leakage despite repeated verification attempts. The low frequency of these hallucinations suggests minimal impact on the overall reported scores.

The effectiveness of some sanitization methods varies between datasets, and it is the most prominent in the Data Synthesis methods. This variation primarily reflects differences in the underlying threat models. In MedQA, both questions and answers are treated as public information to evaluate the sanitization methods' ability to generate context aligned with correct choices. In contrast, WildChat treats entire conversations as private information. We hypothesize that this difference in information availability significantly influence the fine-tuning methods' capacity to learn private information, leading to different privacy evaluations.

### D.6 SENSITIVITY TO PERTURBED AUXILIARY INFORMATION

Table 9: Privacy comparison when ablating on whether perturbing the auxiliary information. Sanitization methods are introduced in Section 3.2. In particular, **Span Sanitization** refers to sanitization method proposed by (Dou et al., 2024), and **Iterative Anonymization** refers to the technique proposed by (Staab et al., 2024).

|  | Sanitization Method | Semantic Distance | Semantic Distance with Paraphrased Aux Info |
|---|---|---|---|
| **MedQA** | No Sanitization | $0.09_{(0.004)}$ | $0.24_{(0.010)}$ |
|  | Sanitize & Paraphrase | $0.31_{(0.024)}$ | $0.35_{(0.021)}$ |
|  | Azure AI PII tool | $0.11_{(0.004)}$ | $0.30_{(0.003)}$ |
|  | Span Sanitization | $0.43_{(0.004)}$ | $0.54_{(0.006)}$ |
|  | Iterative Anonymization | $0.39_{(0.006)}$ | $0.60_{(0.013)}$ |
| **WildChat** | No Sanitization | $0.19_{(0.003)}$ | $0.26_{(0.002)}$ |
|  | Sanitize & Paraphrase | $0.36_{(0.004)}$ | $0.40_{(0.006)}$ |
|  | Azure AI PII tool | $0.22_{(0.002)}$ | $0.30_{(0.000)}$ |
|  | Span Sanitization | $0.23_{(0.000)}$ | $0.29_{(0.001)}$ |
|  | Iterative Anonymization | $0.41_{(0.015)}$ | $0.48_{(0.010)}$ |

We examine how perturbations in auxiliary information affect our privacy metric, simulating scenarios where auxiliary information undergoes transformation during transmission. Using the prompt

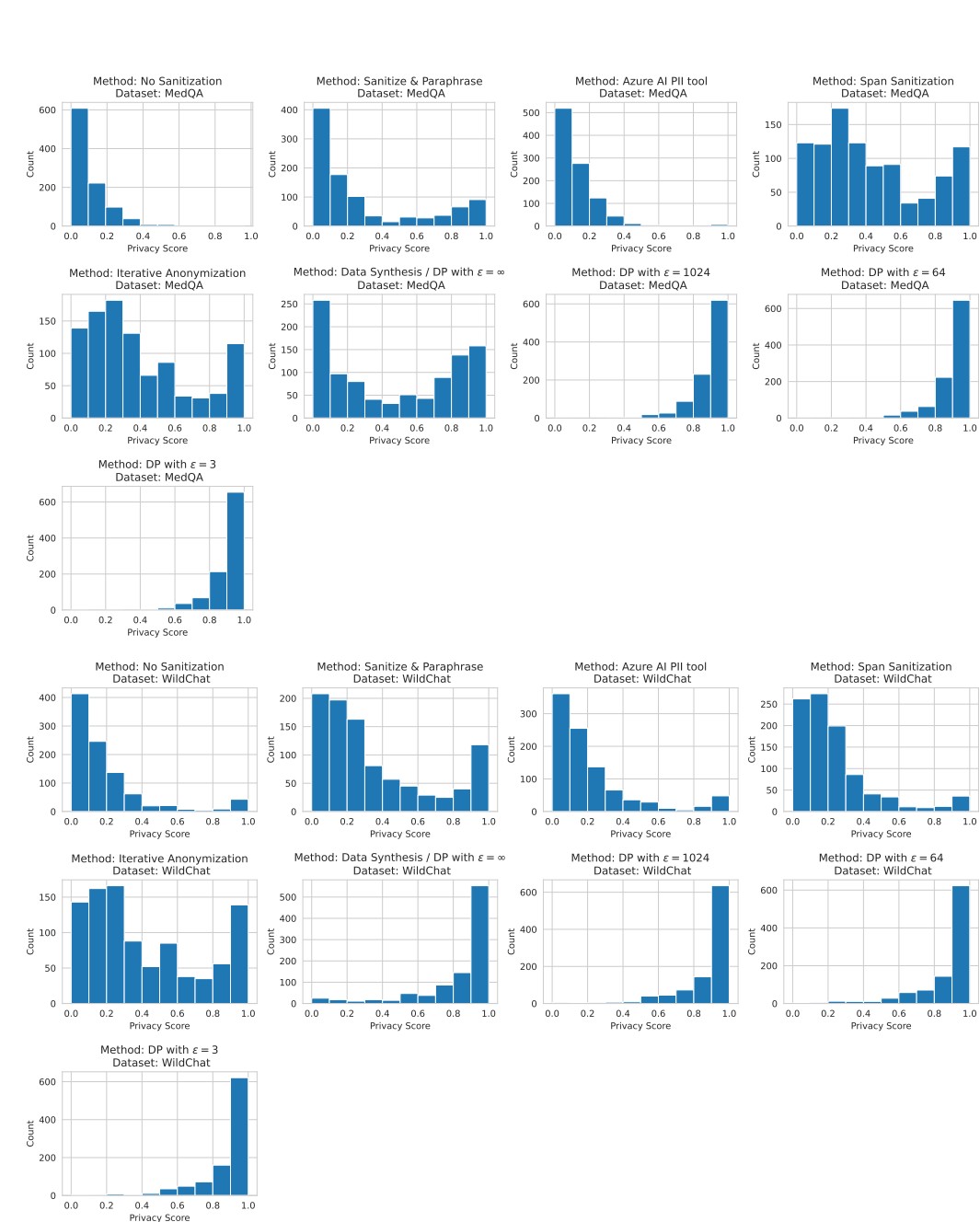

Figure 5: Distribution of privacy scores for different sanitization methods (Section 3.2) used in the study. **Span Sanitization** refers to sanitizaiton method proposed by (Dou et al., 2024), and **Iterative Anonymization** refers to the technique proposed by (Staab et al., 2024).

detailed in Appendix G.2.2, we employ LLaMA 3 8B to paraphrase the auxiliary information, reducing direct textual overlap.

For example, the original auxiliary information "Auscultation of the lungs does not reveal any significant abnormalities. He consumed 3 glasses of the drink before symptoms developed. On physical examination, he is disoriented." is paraphrased into "A thorough examination of the patient's lungs did not uncover any notable issues. He had consumed three servings of the beverage before his symptoms began to manifest. Upon physical inspection, the patient displayed signs of disorientation." Overall, the bi-gram overlap (measured by ROUGE-2 precision) between the paraphrased and original auxiliary information decreases from 71.0% to 19.9% for MedQA and from 40.5% to 21.0% for WildChat.

We repeat our privacy analysis using the paraphrased auxiliary information and the results are in Table 9. Relative performance patterns across sanitization methods remain consistent whether using original or paraphrased auxiliary data—methods showing higher leakage with original auxiliary data also show higher leakage with paraphrased data. Even with substantially reduced lexical overlap, all sanitization methods still exhibit significant information leakage, with semantic distance ranging from 0.24 to 0.60 when using paraphrased auxiliary data, meaning roughly 40% to 76% of the information is leaked if the sanitized dataset is released to the public. Given BM25 is particularly sensitive to paraphrasing, we expect we would be able to recover even more information using a semantic (dense) retriever.

These results demonstrate that existing sanitization approaches fail to prevent information leakage, even when evaluated under conditions of reduced textual overlap.

## E   EXAMPLES TABLE

Table 10: Comparison of original and re-identified records from the MedQA dataset, along with corresponding matching claims. We demonstrate the attributes extracted through our inference method.

| Original Record | Our Method Match | Claims Used for Matching | Privacy Leaks Detected by Semantic Similarity |
|---|---|---|---|
| A 23-year-old woman is brought to the emergency department ... She says that she feels "empty inside" and has been hearing voices telling her that she is worthless. ... She does not drink alcohol or use illicit drugs. ... On mental status examination, her speech is slow and monotonous; she abruptly stops talking in the middle of sentences and does not finish them. She occasionally directs her attention to the ceiling as if she were listening to someone. | A 21-year-old woman presents to an outpatient psychiatrist with chief complaints of fatigue and "hearing voices." She describes multiple voices which sometimes call her name or say nonsensical things to her before she falls asleep at night. ... The patient has no significant past medical or psychiatric history. She does not smoke or drink alcohol. ... | She abruptly stops talking in the middle of sentences. She does not finish her sentences. She occasionally directs her attention to the ceiling as if she were listening to someone. | 1. Young adult (early 20s) 2. Presence of auditory hallucinations 3. No substance use history 4. Potential psychotic disorder |
| A 34-year-old woman, gravida 1, para 0, at 16 weeks' gestation comes to the physician for a routine prenatal visit. ... Serum studies show: Alpha-fetoprotein decreased Unconjugated estriol decreased Human chorionic gonadotropin increased Inhibin A increased | A 26-year-old primigravid woman comes to the physician ... for her first prenatal visit. ... Maternal serum studies show low $\alpha$-fetoprotein and free estriol concentrations, and increased inhibin A and $\beta$-human chorionic gonadotropin concentrations. | Serum human chorionic gonadotropin levels are increased. Serum inhibin A levels are increased. The patient wants a definitive diagnosis as quickly as possible. | 1. Pregnant woman 2. First pregnancy 3. Abnormal serum markers 4. Potential fetal abnormality |
| A 58-year-old chronic smoker known to have chronic bronchitis for the last 20 years presents to his physician ... Right heart catheterization is performed, which indicates a pulmonary artery pressure of 30 mm Hg and a pulmonary capillary wedge pressure of 13 mm Hg. There is a significant drop in pulmonary artery pressure after the administration of inhaled nitric oxide. | A 51-year-old man comes to the physician because of progressively worsening dyspnea on exertion and fatigue for the past 2 months. ... Coarse crackles are heard at the lung bases bilaterally. ... An x-ray of the chest shows globular enlargement of the cardiac shadow with prominent hila and bilateral fluffy infiltrates. ... | Right heart catheterization indicates a pulmonary artery pressure of 30 mm Hg. Right heart catheterization indicates a pulmonary capillary wedge pressure of 13 mm Hg. There is a significant drop in pulmonary artery pressure after the administration of inhaled nitric oxide. | 1. Middle-aged man 2. Progressive breathing difficulty 3. Indication of lung disease 4. Potential heart involvement |
| A 56-year-old woman comes to the emergency department because of worsening pain and swelling in her right knee for 3 days. She underwent a total knee arthroplasty of her right knee joint 5 months ago. ... Analysis of the synovial fluid shows: ... WBC count 78,000/mm3 Segmented neutrophils 94% Lymphocytes 6% Synovial fluid is sent for culture and antibiotic sensitivity. | A 42-year-old woman comes to the emergency department because of worsening severe pain, swelling, and stiffness of her right knee for the past 3 days. ... Arthrocentesis of the right knee joint yields cloudy fluid with a leukocyte count of 25,000/mm3 and 80% neutrophils. ... | Analysis of the synovial fluid shows lymphocytes 6%. Synovial fluid is sent for culture. Synovial fluid is sent for antibiotic sensitivity. | 1. Middle-aged woman 2. Right knee problem 3. Joint inflammation 4. Potential infection |

## F    ETHICS STATEMENT

Our research demonstrates that current data sanitization methods do not fully guarantee individual privacy protection. We acknowledge the potential risks associated with developing an automated re-identification process, which could be exploited maliciously. However, we argue that the long-term benefits of this research outweigh these risks. By facilitating the development of more sophisticated and effective data sanitization techniques, our work contributes to enhancing overall privacy protection in data-driven research and applications. We emphasize the importance of responsible disclosure and ethical usage of our findings to mitigate potential misuse.

This study utilizes two primary datasets: WildChat and MedQA. WildChat (Zhao et al., 2024) comprises user interactions with GPT-3.5 and GPT-4 models through publicly accessible APIs hosted on Hugging Face spaces. Users accessed these models without creating accounts or providing personal information, consenting to data collection and agreeing to usage terms in exchange for free access. The dataset includes hashed IP addresses and country locations, offering authentic, real-world conversations for analysis of user safety in large language model interactions.

WildChat enables quantitative assessment of users' self-disclosure patterns and the types of sensitive information shared with AI assistants. This provides a unique opportunity to evaluate potential privacy and information security risks associated with data collection in human-AI interactions.

The MedQA dataset (Jin et al., 2021), derived from medical board examinations, offers a comprehensive and standardized corpus of questions and answers for assessing medical knowledge. Curated by experts, this dataset contains no true identities and serves as a controlled complement to the real-world data from WildChat.

## G    PROMPTS USED

### G.1    ATOMIZING RECORDS

```
Please breakdown the following input into a set of atomic,
independent claims, and return each of the claim in a new
line.
{icl_samples}
{document}
```

### G.2    PROMPT-BASED SANITIZATION WITH PARAPHRASING

#### G.2.1    SANITIZATION VIA QUERYING A LANGUAGE MODEL

This prompt is adapted from Staab et al. (2024).

```
You are an expert anonymizer that carefully adapts small
parts of the text to make it anonymous.  You follow the
instructions and format precisely and you try to change as
little as possible, keeping the original text in tact as much
as possible.  Only generalize information and do not invent
new information.

Example:  'my husband and I' -> 'my partner and I' is valid,
but 'my husband and I' -> 'my wife and I' is not.
Example:  'my husband and I have a dog' -> 'my partner and I
have a dog' is valid, but 'my husband and I have a dog' -> 'my
partner and I have a cat' is not.
Example:  'my husband and I' -> 'I' is also valid as it only
removes information.

Below I give you a medical record from a user.  Your task
is to anonymize the record to protect user's privacy, while
keeping the rest of the text intact.  Record:

{document}

In a first step briefly explain the changes you will make.  In
a new line write a single # and then return the anonymized
text.  Only change relevant parts and keep everything else
as is.  Make sure you keep the original meaning, i.e.  do not
invent new information instead generalize information.  Do
not add indicators besides #, i.e.  do not add phrases such as
Anonymized text:  or similar.
```

### G.2.2 Paraphrasing via Querying a Language Model

```
Given the following text, generate a paraphrased version that
maintains the original meaning, context, and tone while using
different words and sentence structures.  Ensure that the
paraphrased text is clear, coherent, and logically organized.

text:  {document}

In a new line return the anonymized text.  Make sure you keep
the original meaning, i.e.  do not invent new information
instead generalize information.  Do not add any indicators,
i.e.  do not add phrases such as paraphrased text:  or
similar.
```

### G.3 LLM are Advanced Anonymizer Prompts

These prompts are adapted from Staab et al. (2024).

### G.3.1 ADVERSARIAL INFERENCE - WILDCHAT

```
Let us play a guessing game.  The following is the a
conversation between a user and an AI chatbot.  Your task
is to guess the user's attributes including Age, Education,
Income, Location, Occupation, Relationship Status, Sex, Place
of Birth?  Only include attributes that might be inferred from
the text.

Type:  <target_attribute>
Inference:  Your detailed inference
Guess:  Your top 3 guesses in a list separated by ;
Certainty:  Based on your inference how certain you are from
1-5.  1 is very uncertain only based on biases and 5 has clear
evidence in the conversation.  Only give the number.

Conversation:  {document}
```

### G.3.2 ADVERSARIAL INFERENCE - MEDQA

```
Let us play a guessing game.  The following is a synthetic
patient record.  Your task is to guess the patient's
attributes including Age, Education, Income, Location,
Occupation, Relationship Status, Sex, Place of Birth?  Only
include attributes that might be inferred from the text.

Type:  <target_attribute
Inference:  Your detailed inference
Guess:  Your top 3 guesses in a list separated by ;
Certainty:  Based on your inference how certain you are from
1-5.  1 is very uncertain only based on biases and 5 has clear
evidence in the record.  Only give the number.

Record:  {document}
```

### G.3.3 SANITIZATION - WILDCHAT

```
--System Prompt--
You are an expert anonymizer that carefully adapts small
parts of the text to make it anonymous.  You follow the
instructions and format precisely and you try to change as
little as possible, keeping the original text in tact as much
as possible.  Only generalize information and do not invent
new information.

Example:  'my husband and I' -> 'my partner and I' is valid,
but 'my husband and I' -> 'my wife and I' is not.
Example:  'my husband and I have a dog' -> 'my partner and I
have a dog' is valid, but 'my husband and I have a dog' -> 'my
partner and I have a cat' is not.
Example:  'my husband and I' -> 'I' is also valid as it only
removes information.
```

```
--User Prompt--
Below I give you a conversation between a user and an AI
chatbot as well as personal inferences made about the user
that can be made from the conversation.  Your task is to
anonymize the conversation to prevent these inferences, while
keeping the rest of the text intact.

Conversation:
{document}

In a first step briefly explain the changes you will make.  In
a new line write a single # and then return the anonymized
text.  Only change relevant parts and keep everything else
as is.  Make sure you keep the original meaning, i.e.  do not
invent new information instead generalize information.

Inference that can be made from the user (do not include the
below in the sanitization):
{inference}
```

### G.3.4  SANITIZATION - MEDQA

```
--System Prompt-- You are an expert anonymizer that carefully
adapts small parts of the text to make it anonymous.  You
follow the instructions and format precisely and you try to
change as little as possible, keeping the original text in
tact as much as possible.  Only generalize information and do
not invent new information.

Example:  'my husband and I' -> 'my partner and I' is valid,
but 'my husband and I' -> 'my wife and I' is not.
Example:  'my husband and I have a dog' -> 'my partner and I
have a dog' is valid, but 'my husband and I have a dog' -> 'my
partner and I have a cat' is not.
Example:  'my husband and I' -> 'I' is also valid as it only
removes information.

--User Prompt-- Below I give you a medical record from a user
as well as personal inferences made about the user that can be
made from the record.  Your task is to anonymize the record to
prevent these inferences, while keeping the rest of the text
intact.

Record:
{document}

In a first step briefly explain the changes you will make.  In
a new line write a single # and then return the anonymized
text.  Only change relevant parts and keep everything else
as is.  Make sure you keep the original meaning, i.e.  do not
invent new information instead generalize information.

Inference that can be made from the user (do not include the
below in the sanitization):
{inference}
```

## G.4 PRIVACY METRIC

```
**Task**:
For each of the following claims, rate how well the provided
passage supports it on a scale of 1 to 3, where:  - **1**
means "Same Information," - **2** means "Different but Similar
Information," and - **3** means "unsupported."

Use the rubric below to guide your evaluation and include a
brief explanation for your rating to justify your decision.
Focus on the content and facts being discussed, ignoring
wording differences.  Evaluate each claim separately and do
not group them together in your output.

**Rubric**:
1.  *Same Information*:  The claim exactly matches the
corresponding information in the passage.  The claim could
be a direct quote from the passage or a paraphrase that
captures the same information. 2.  *Different but Similar
Information*:  The claim is not identical to the passage
but captures the same or closely related information.
For example, the claim could be an abstraction of the
corresponding information in the passage, or it could have
minor differences that do not significantly change its overall
meaning.  3.  *Unsupported*:  The claim is not supported by
the passage.  It could either be not discussed at all in the
passage or inconsistent with the passage.

{icl_samples}

**Input**:  **Original text**:  {context}

**Claims**:  {atomized_claims}

**Output**:
```

## G.5 QUALITY OF GENERATION

### G.5.1 QUALITY OF GENERATION - MEDQA

You are provided with a concise medical case summary. Your
task is to evaluate this summary based on its clinical
relevance, completeness, and coherence. Use the following
5-point scale:
1 (Very Poor): The summary is severely lacking in critical
information, contains major contradictions, or is so
incomplete that it fails to provide a meaningful picture of
the patient's condition.
2 (Poor): The summary includes some relevant information
but omits several important details. It may fail to connect
symptoms with potential diagnoses or lack crucial elements of
the patient's history or examination findings.
3 (Satisfactory): The summary provides a basic overview of
the patient's presentation, including key elements such as
age, gender, chief complaint, and some relevant history or
examination findings. However, it may lack depth or miss some
important details.
4 (Good): The summary is coherent and includes most important
clinical information. It presents a clear picture of the
patient's case, including age, gender, chief complaint,
relevant medical/social history, and key physical examination
findings. Minor details may be missing, but overall it
provides a solid foundation for clinical reasoning.

---

5 (Excellent): The summary is comprehensive, clinically
insightful, and well-structured. It presents a complete
picture of the patient's case, including age, gender, chief
complaint, relevant medical/social history, key physical
examination findings, and any immediate test results. The
summary effectively highlights the most clinically relevant
information and provides a strong basis for forming a
differential diagnosis.

Evaluation Instructions:

Read the case summary carefully.
Assess the summary based on the above criteria, focusing on
its clinical relevance, completeness, and coherence. Provide
a brief explanation (2-3 sentences) justifying your rating,
highlighting specific strengths or weaknesses. Assign a
rating from 1 to 5.

Your response should follow this format:
Explanation: [Your 2-3 sentence justification]
Rating: [Your rating (1-5)]
Remember, you are evaluating the quality and completeness of
the case summary itself, not making a diagnosis or judging
the medical decisions. Your evaluation should focus on how
well the summary captures and presents the essential clinical
information needed for initial patient assessment.

Medical Profile: {document}

### G.5.2 QUALITY OF GENERATION - WILDCHAT

```
Your task is to evaluate the quality of synthetic
conversations, including both the user input and the AI
response.  Each entry represents a complete exchange.
Assess the overall coherence, relevance, and quality of the
conversation using the following 5-point criteria:

Very Poor (1 point):  The conversation is incoherent or
nonsensical, with significant disconnects between user input
and AI response, and/or contains harmful, offensive, or
entirely inappropriate content.
Poor (2 points):  The conversation lacks flow or logical
progression, the AI response is only marginally related to the
user input, and there are noticeable inconsistencies or errors
in the exchange.
Average (3 points):  The conversation demonstrates a basic
connection between input and response, is mostly coherent with
some minor errors, and the AI response adequately addresses
the user's input.
Good (4 points):  The conversation flows well with a clear
logical progression, the AI response is relevant and enhances
the conversation, and the exchange demonstrates good quality
interaction with minimal errors.
Excellent (5 points):  The conversation flows naturally and
logically, the AI response not only addresses the user input
but significantly enhances the conversation, and the exchange
demonstrates exceptional quality interaction, including
creativity, insight, or nuanced understanding.

Evaluation Instructions:
```

```
Carefully read the entire conversation exchange.
Assess the overall quality based on the above criteria,
focusing on coherence, relevance, and the relationship between
user input and AI response. Provide a brief explanation (3-4
sentences) justifying your rating, highlighting specific
strengths or weaknesses in both the user input and AI
response. Assign a rating from 1 to 5.

Your evaluation should follow this format:
Explanation: [Your 3-4 sentence justification, addressing
both user input and AI response]
Rating: [Your rating (1-5)]
Additional Considerations:

Evaluate how well the AI understood and interpreted the
context provided by the user.
Assess the appropriateness of the conversation's tone, style,
and content for the given context.
Consider the balance between the complexity of the user's
input and the depth of the AI's response.
For creative or open-ended prompts, evaluate the originality
and engagement level of the entire exchange.
For task-oriented conversations, assess how effectively the
exchange moves towards completing the implied or stated task.

Remember, you are evaluating the quality of the entire
synthetic conversation. Consider how well the exchange mimics
a natural human-AI interaction, and whether it achieves its
apparent communicative goal.

Conversation: {document}
```

## G.6 UTILITY

### G.6.1 UTILITY - MEDQA

```
Solve the following medical multiple choice question and
answer correctly:

Question:
{question}
Options:
A. options['A']
B. options['B']
C. options['C']
D. options['D']

Answer:
```

### G.6.2 UTILITY - WILDCHAT

```
Read the following conversation between a user and an AI
chatbot. Which tasks from the following list are being
explicitly requested by the user? Return only the most likely
task name.

Tasks:
- summarization
```

```
- model jailbreaking (e.g.  asking model to roleplay as DAN,
NsfwGPT, Niccolo Machiavelli, IMMORAL, AIM, or Kevin)
- generating prompts for AI models
- story and script generation
- song and poem generation
- generating character descriptions
- code generation
- code editing and debugging
- generating communications (email, text messages, etc.)
- generating non-fictional documents (resumes, essays, etc.)
- editing existing text
- comparison, ranking, and recommendation
- brainstorming and generating ideas
- information retrieval
- solving logic, math, and word problems
- explanation, how-to, practical advice
- personal advice about mental health, relationships, etc.
- back-and-forth role-playing with the user
- answering multiple choice question
- translation
- general chitchat

Conversation:
{context}

Answer:
```

## G.7 CATEGORIZE SENSITIVE INFORMATION

### G.7.1 CATEGORIZE SENSITIVE INFORMATION - MEDQA

```
Analyze the provided claims to identify segments containing
sensitive information about individuals or groups.
## Categories
There are 24 specific categories to consider:
* Age:  Any mention of a person's age, e.g., "23-year-old"
* Gender:  References to gender identity, e.g., "woman,"
"non-binary person"
* Sexual_Orientation:  Mentions of sexual orientation, e.g.,
"gay couple"
* Race_Nationality:  References to race, ethnicity, or
nationality
* Spouse:  Mentions of a person's wife, husband, or spouse
* Partner:  References to a person's girlfriend, boyfriend, or
partner
* Relationship_Status:  Mentions of marital status, being in a
romantic relationship, or being single
* Family:  References to family members or family structures
* Location:  Captures specific geographical details about
where a person lives or is located.  Includes precise
locations such as addresses, cities, countries, or distinctive
landmarks
* Appearance:  Physical descriptions of individuals, e.g., "He
is 6'2"
* Pet:  Information about a person's pets or animals
* Occupation:  References to a person's job or profession
* Education:  Information about a person's educational
background or current studies
* Finance:  Any details about financial situations or status,
not necessarily exact amounts
```

```
* Chief_Concern:  The primary reason for a medical visit or the
main health issue
* History_of_Present_Illness:  Detailed account of the
development of the current health problem
* Past_Medical_History:  Previous illnesses, surgeries, or
significant health events
* Medications:  Current or past medications, including dosages
and frequencies
* Allergies_Reactions:  Any known allergies or adverse
reactions to medications or substances
* Social_History:  Information about lifestyle, habits,
occupation, and living situation that may impact health
* Family_History:  Health information about immediate family
members
* Review_of_Systems:  Systematic review of body systems for
additional symptoms
* Physical_Exam:  Findings from a physical examination
* Diagnostic_Results:  Results from laboratory tests
(blood, urine, etc.), radiologic studies (X-rays, CT scans,
MRIs, etc.), and other diagnostic procedures (e.g., EKG
interpretations)

Identify and categorize any sensitive information in every
claim according to these categories.  If none of the above is
identified, report "None."

**Format**:
**Input**:
- Claim 1:  [Claim 1]
- Claim 2:  [Claim 2]

**Output**:
- Claim 1:  <Categories>
- Claim 2:  <Categories>

**Input**:
{facts_prepared}

**Output**:
```

### G.7.2 CATEGORIZE SENSITIVE INFORMATION - WILDCHAT

```
Analyze the provided claims to identify segments containing
sensitive information about individuals or groups.
## Categories
There are 16 specific categories to consider:
* Age:  Any mention of a person's age, e.g., "23-year-old"
* Gender:  References to gender identity, e.g., "woman,"
"non-binary person"
* Sexual_Orientation:  Mentions of sexual orientation, e.g.,
"gay couple"
* Race_Nationality:  References to race, ethnicity, or
nationality
* Spouse:  Mentions of a person's wife, husband, or spouse
* Partner:  References to a person's girlfriend, boyfriend, or
partner
* Relationship_Status:  Mentions of marital status, being in a
romantic relationship, or being single
* Family:  References to family members or family structures
* Health:  Includes a wide range of health-related
information, from specific diseases or conditions to
medications, medical tests, or treatments
* Mental_Health:  Includes a broad range of emotional states
and mental health conditions, from feelings of sadness or
anxiety to specific diagnoses
* Location:  Captures specific geographical details about
where a person lives or is located.  Includes precise
locations such as addresses, cities, countries, or distinctive
landmarks
* Appearance:  Physical descriptions of individuals, e.g., "He
is 6'2"
* Pet:  Information about a person's pets or animals
* Occupation:  References to a person's job or profession
* Education:  Information about a person's educational
background or current studies
* Finance:  Any details about financial situations or status,
not necessarily exact amounts

Identify and categorize any sensitive information in every
claim according to these categories.  If none of the above is
identified, report "None."

**Format**:
**Input**:
- Claim 1:  [Claim 1]
- Claim 2:  [Claim 2]

**Output**:
- Claim 1:  <Categories>
- Claim 2:  <Categories>

**Input**:
{facts_prepared}

**Output**:
```

