# OpenReview forum: "A False Sense of Privacy: Evaluating Textual Data Sanitization Beyond Surface-level Privacy Leakage"
_ICLR.cc/2025/Workshop/BuildingTrust — BuildingTrust_

### Official Review · Reviewer_xepz · 2025-03-02
**Good paper on an important topic, but missing a constructive angle.**

**Rating:** 6
**Confidence:** 5

**Review:**

The paper conducts a thorough evaluation of various text anonymization methods under a de-anonymization setting where side information is available to the adversary. This is a strictly stronger setting than in previous works (e.g., [1,2]), which is an important and interesting extension of the evaluation of the heuristic anonymization methods of PII-removal, and span- or text-sanythization, bringing them closer in setting to the threat model of DP. As such, I believe the paper is a good addition to the workshop.

However, there are certain limitations I would like to highlight, which, if addressed by the authors in a future revision of the paper, could potentially increase the paper's impact:

- Currently, the paper does not offer clear insights beyond what has already been known, it simply, less-interestingly so, shows that they also hold in the examined setting. Namely, the paper's key insights are: (i) DP trades utility for privacy; (ii) heuristic methods (sanythization) provide weaker privacy than DP; and (iii) PII removal does not work once the information is not surface-level. However, I believe there could be more interesting conclusions be drawn from the setting; e.g., for instance by further analyzing the type and impact of the side information and establishing concrete realistic settings where the anonymizing party does not have access to this information but the adversary does.

- This leads over to my second point. I think the current evaluation of certain methods misses a key point: giving access to the side information also in the anonymization method. For instance, expanding the set of attributes to anonymize using [2] to the ones detected in the text, and also giving the iterative anonymizer the side information should show significant performance increases. While the authors could argue against how realistic such an experiment would be, this (i) first has to be justified in the threat model (see my point above), and (ii) could still showcase the full potential of each method, making sure that the lack of performance does not stem from simply the mismatch between the current evaluation setting and the one that was originally used when designing these methods.

- I would also like to have a more fine-grained analysis in the experiments of where the final information leakage comes from, i.e., is it information that is still contained in the sanythized text, is it information that is only contained in the side information, or does come to make sense only once these two texts are combined. This is important as, at the moment it is not clear what use the side information has in the whole process, particularly as the privacy metric is also calculated w.r.t. to the private information in the original text and not some underlying private profile. This leads to vastly different types of leakages being masked by the metric, compare for instance the following two cases: (i) the type of leakage displayed also in the intro figure, where the side information has no role in the final private information that was extracted as all the information was still contained in the sanythized text; and (ii) e.g., some feature is present in both the original text and in the side information (e.g., age) and the two can be linked based on a seemingly irrelevant, non-private feature; then, even a perfect sanythization of the original text (w.r.t. the set of private features) will enable the reconstruction of the age feature in the current setting.

- Finally, I think the paper currently lacks some clear outlook, takeaway, or next steps on which one could build. At the moment, it simply makes the statement that certain methods don't work at all, certain methods are too weak, and that DP makes the text useless. But there is no further analysis going into which directions would be promising to explore, or even, what would be a best-practice approach to anonymization given the current tools. I think this severely limits the potential impact of the paper, as it reads more like an evaluation report than something that one could build upon.


**References**

[1] Y Dou et al., Reducing Privacy Risks in Online Self-Disclosures with Language Models. ACL 2024.

[2] R Staab et al. Language Models are Advanced Anonymizers. ICLR 2025.

---

### Official Review · Reviewer_fCTP · 2025-03-02
**The work highlights the inadequacies of current text data sanitization methods, introducing a novel dataset-level privacy metric and holistic evaluation framework, but is limited by an ambiguous definition of privacy and a restricted number of datasets**

**Rating:** 6
**Confidence:** 3

**Review:**

A study of text data privacy shows that current sanitisation approaches are inadequate. Current PII removal methods leave 89% of information vulnerable, and even synthetic data without differential privacy remains exploitable. While DP synthesis offers better protection, it comes at a significant performance cost.

Pros:
- This work introduces a dataset-level privacy metric that addresses the limitations of current data sanitisation methods.
- The authors provide a comprehensive framework with a holistic evaluation of the trade-offs in different sanitisation techniques.

Disadvantages:
- Ambiguous definition of privacy, which poses a significant challenge in developing universally applicable privacy metrics and protection methods.
- Limited number of datasets - it would be beneficial to extend the research with additional datasets from other domains.

---

### Official Review · Reviewer_xgvo · 2025-03-02
**Paper Review**

**Rating:** 7
**Confidence:** 4

**Review:**

> Summary

​In this study, the authors conduct a comprehensive evaluation of privacy leakage associated with data sanitization methods. By employing a re-identification attack model and a semantic-based privacy metric, their approach more effectively captures privacy risks compared to traditional lexical matching techniques. The experiments are thorough and supportive.

> Pros:

1.The topic investigated is interesting, and the findings are engaging.

2.The experimental results are comprehensive.

> Cons:

1.A necessary assumption for the attacker is that they must have access to relevant auxiliary information about the subject, which may not always be easily obtainable, especially in privacy-sensitive domains.

2.I was particularly curious about the results showing that even with $\epsilon  = 1024$, the privacy metric increased significantly, while utility metrics dropped substantially. It would be helpful if the authors also reported the clip norm and noise scale used in the experiments.

---

### Decision · Program_Chairs · 2025-03-04

Accept